# Mechanism-Based Approach to New Antibiotic Producers Screening among Actinomycetes in the Course of the Citizen Science Project

**DOI:** 10.3390/antibiotics11091198

**Published:** 2022-09-05

**Authors:** Inna A. Volynkina, Yuliya V. Zakalyukina, Vera A. Alferova, Albina R. Belik, Daria K. Yagoda, Arina A. Nikandrova, Yuliya A. Buyuklyan, Andrei V. Udalov, Evgenii V. Golovin, Maxim A. Kryakvin, Dmitrii A. Lukianov, Mikhail V. Biryukov, Petr V. Sergiev, Olga A. Dontsova, Ilya A. Osterman

**Affiliations:** 1Center of Life Sciences, Skolkovo Institute of Science and Technology, Bolshoy Boulevard 30, bld. 1, 121205 Moscow, Russia; 2Department of Chemistry, Lomonosov Moscow State University, Leninskie Gory 1, 119991 Moscow, Russia; 3Center for Translational Medicine, Sirius University of Science and Technology, Olympic Avenue 1, 354340 Sochi, Russia; 4Department of Soil Science, Lomonosov Moscow State University, Leninskie Gory 1, 119991 Moscow, Russia; 5Shemyakin-Ovchinnikov Institute of Bioorganic Chemistry, Miklukho-Maklaya 16/10, 117997 Moscow, Russia; 6Gause Institute of New Antibiotics, B. Pirogovskaya 11, 119021 Moscow, Russia; 7School of Bioengineering and Bioinformatics, Lomonosov Moscow State University, Leninskie Gory 1, 119991 Moscow, Russia; 8Department of Biology, Lomonosov Moscow State University, Leninskie Gory 1, 119991 Moscow, Russia

**Keywords:** citizen science, crowdsourcing, antibiotic producers screening, actinomycetes, reporter systems, chartreusin, pikromycin

## Abstract

Since the discovery of streptomycin, actinomycetes have been a useful source for new antibiotics, but there have been diminishing rates of new finds since the 1960s. The decreasing probability of identifying new active agents led to reduced interest in soil bacteria as a source for new antibiotics. At the same time, actinomycetes remain a promising reservoir for new active molecules. In this work, we present several reporter plasmids encoding visible fluorescent protein genes. These plasmids provide primary information about the action mechanism of antimicrobial agents at an early stage of screening. The reporters and the pipeline described have been optimized and designed to employ citizen scientists without specialized skills or equipment with the aim of essentially crowdsourcing the search for new antibiotic producers in the vast natural reservoir of soil bacteria. The combination of mechanism-based approaches and citizen science has proved its effectiveness in practice, revealing a significant increase in the screening rate. As a proof of concept, two new strains, *Streptomyces* sp. KB-1 and BV113, were found to produce the antibiotics pikromycin and chartreusin, respectively, demonstrating the efficiency of the pipeline.

## 1. Introduction

The spread of antibiotic resistance is one of the major problems for modern antibacterial therapy. The most serious threats are methicillin-resistant *Staphylococcus aureus* (MRSA), vancomycin-resistant *Enterococcus* (VRE), multidrug-resistant (MDR) *Acinetobacter baumannii* and β-lactam-resistant *Enterobacteriaceae* [1]. The discovery of new antibacterial compounds may help to solve the problem, and soil bacteria still remain one of the most promising sources of bioactive natural products [2,3].

At the same time, traditional screening of soil actinomycetes for the production of antimicrobial agents often yields known molecules [4]. For this reason, new strategies for the discovery of new antibiotics have been developed [5,6,7,8]. One possible solution to this so-called “rediscovery” issue is to collect samples from novel ecological niches and to isolate strains that occur there. Specifically, extreme environments such as those with high salinity or alkalinity are a promising reservoir of new antimicrobial producers [9]. Moreover, “previously uncultured” microorganisms, that do not readily grow under laboratory conditions, may also serve as an untapped source of secondary metabolites [10]. The advances in cultivation techniques led to the discovery of a new antibiotic, teixobactin, which inhibits cell wall biosynthesis [11]. Metagenomic libraries with DNA fragments extracted directly from soil and cloned in appropriate vectors, which are expressed in a culturable bacterium, enable researches to screen genes from “unculturable” microorganisms for antimicrobial production [12,13]. Furthermore, whole-bacterial genome sequencing and bioinformatics tools for genome mining make it possible to identify biosynthetic gene clusters, even those predicted to encode novel antibiotics [14,15]. However, activation of transcriptionally silent gene clusters is still a technical challenge.

Whereas some researchers are focused on developing new technically sophisticated, usually expensive and even more time-consuming approaches, including genome mining, bioinformatics analysis of omics data, screening of synthetic chemical libraries or target-based screening, others suggest expanding antimicrobial discovery activity through crowdsourcing [16]. A widely known innovative program named the Small World Initiative (SWI, www.smallworldinitiative.org (accessed on 25 August 2022)) was formed at Yale University in 2012. As part of the citizen science project, students from around 150 participating schools isolated bacteria from their local environments, determined antibiotic production among those strains and extracted active compounds for further characterization. Later, SWI presented an alternative pipeline, based on identification of biosynthetic gene clusters responsible for antimicrobial production phenotype, using transposon mutagenesis [17]. All the experiments were carried out in well-equipped research laboratories. Inspired by the SWI, in 2015, the Microbiology Society implemented a spin-off program, Antibiotics Unearthed, in the UK (www.microbiologysociety.org (accessed on 27 August 2022)). During the project, selected school or college students, as well as undergraduate students, analyzed their soil samples for antibacterial compounds and investigated any potential compounds found. Moreover, the Microbiology Society held a series of events, giving the general public an opportunity to collect a soil sample and prepare it for scientific analysis during their visit. There are other citizen science projects aimed at crowdsourcing only soil sampling: Citizen Science Soil Collection Program (University of Oklahoma, since 2010, www.whatsinyourbackyard.org (accessed on 26 August 2022)), Swab and Send (since 2015, [16]) and Drugs From Dirt (Rockefeller University, since 2016, www.drugsfromdirt.org (accessed on 25 August 2022)). A similar Citizen Science project is currently being carried out in Russia starting from 2021. In this project, we decided to crowdsource not only sampling but also some primary experiments such as isolation of single colonies and testing them against indicator reporter strains. Therefore, we have designed an optimized workflow that enables school students and the interested public to perform scientific experiments in educational laboratories which are not specially equipped, or even at home.

One way to address the “rediscovery” issue is to determine the antimicrobial target and mode of action during the screening. It might help to narrow down the pool of bioactive natural products under consideration, and sometimes it might even help to prematurely identify the compound produced. Mechanism of action and antibiotic target could be revealed by different methods: direct measurement of the main cellular process inhibition by incorporation of labeled metabolites [18], proteomic signature [19], cytological profiling of bacteria using fluorescence microscopy [20], selection of resistant clones followed by genome sequencing [21], genome mining [22] and different reporter systems [23]. We consider reporter systems to be the most preferred tool for the Citizen Science project, since they can provide information at an early stage of screening. In addition, we recently developed such a mechanism-oriented reporter for high-throughput sorting of antibiotic producers by their mode of action [24]. In this study, we present a set of new constructs encoding visible fluorescent proteins or β-galactosidase, which can be detected without special equipment. With these reporter plasmids being transformed into antibiotic-sensitive indicator strains, citizen scientists could conduct a mechanism-oriented screening on their own, in educational laboratories or even at home.

Here, we propose a novel pipeline of the Citizen Science project aimed at finding new antibiotics, starting from a soil sample and ending with an antibiotic structure. The whole process could be divided into two stages: (a) isolation and screening for new producing strains; and (b) identification of active compounds. Crowdsourcing at the first stage of the workflow significantly increases the speed of research and discovery rates. The identification of active compounds (the second stage) should be conducted by a specialist in the research laboratory, because it requires the usage of high-pressure liquid chromatography (HPLC), mass spectrometry, etc. The pipeline described here has been illustrated with two examples: *Streptomyces* sp. KB-1 and BV113 isolates were found to produce the antibiotics pikromycin (translation inhibitor) and chartreusin (SOS response inducer), respectively. The combination of citizen science with the mechanism-based approach and modern analytical methods allows quick and efficient discovery of new antibiotic producers.

## 2. Results

### 2.1. New Reporter Systems for Elucidating the Mode of Antibiotic Action in Application to the Citizen Science Project

Recently we developed a double reporter system, pDualrep2 [24], which consists of two fluorescent protein genes, *turbo-rfp* and *katushka2S* (Figure 1A). The expression of *turbo-rfp* is controlled by the SOS-inducible *sulA* gene promoter [25]. Whereas the expression of *katushka2S* is regulated by the modified *trpL* attenuator sequence [26]. Potent antibiotics, which stall the ribosome on the *trpL2A* open reading frame, prevent premature transcription termination and promote transcription of the entire gene. Previously, this system was successfully applied, resulting in the discovery of new antibiotic producers [27,28]. However, specific equipment is required to distinguish between the fluorescent signals from TurboRFP (553/574 nm) and Katushka2S (588/633 nm) (Figure 1B).

In order to keep the advantage of the dual reporter system pDualrep2, but to avoid the necessity of specific equipment, the *katushka2S* gene was replaced by the β-galactosidase gene (*lacZ*), the expression of which could be visually detected by the blue-colored product in the presence of X-gal substrate (Appendix A). The *turbo-rfp* gene was left unmodified since its expression is clearly visible in orange-red color to the naked eye. The new plasmid was named pDualrep3. In practice, it demonstrated a high background signal due to the nonspecific hydrolysis of X-gal and, in part, leakage of the *trpL2A* regulatory sequence, so that only potent protein synthesis inhibitors (such as chloramphenicol and fusidic acid) could be detected with the pDualrep3 reporter.

This problem and the inability to resolve the situation when one sample induces the expression of both reporter genes, prompted us to design single reporter constructs. Based on fluorescent protein genes *turbo-rfp* and *katushka2S*, we created three new plasmids, named pTrpL2A-RFP, pTrpL2A-Katushka2S and pSulA-RFP, and validated them with a set of antibiotics (Figure 2). Both pTrpL2A-RFP and pTrpL2A-Katushka2S demonstrated strong reporter induction upon treatment with antibiotics that inhibit protein biosynthesis during the elongation step, such as chloramphenicol, puromycin, tetracycline, erythromycin, fusidic acid and lincomycin (Figure 2A,B). Consistent with previous data, spectinomycin and clindamycin demonstrated barely visible reporter induction, while streptomycin and kanamycin showed no induction at all [24]. It is worth considering that only antibiotics that cause ribosome stalling on the *trpL2A* open reading frame could be detected by means of these reporters. For this reason, aminoglycoside antibiotics which predominantly cause mRNA misreading, such as streptomycin and kanamycin, would stay undercover [29]. In addition, we observed that cytotoxic antibiotic doxorubicin, which is known to intercalate DNA and inhibit topoisomerases in bacteria and eukaryotic cells [30,31,32], induced *trpL2A*-containing reporters. We assume that doxorubicin may bind to ribosomal RNA, thus impeding the ribosome in synthesizing polypeptides.

The construct pSulA-RFP demonstrated clear reporter induction upon treatment with doxorubicin and levofloxacin (Figure 2C). Both of them are known to function by inhibiting bacterial DNA gyrase, contributing to DNA damage and subsequent SOS response [30,33]. Moreover, we observed a barely visible reporter induction in the case of rifampicin, which is known to inhibit bacterial RNA polymerase [34].

Thus, plasmids pTrpL2A-Katushka2S and pSulA-RFP were chosen as the most convenient for use in the Citizen Science project. Being transformed into antibiotic-sensitive bacterial strains [35], they may provide primary information about the action mechanism of antimicrobial agents produced by soil actinomycetes. Visually detectable induction of reporter gene expression makes it possible to work with reporter strains in the absence of special equipment, as for example in educational laboratories or at home.

### 2.2. Stocks of Freeze-Dried E. coli Reporter Cells Are Suitable for Application in the Citizen Science Project

As usual, stocks of *E. coli* strains transformed with reporter plasmids are stored in freezers at −80 °C until the experiment. It is almost impossible to equip all educational laboratories with such freezers, not to mention the problem of stocks transportation. In order to work with reporter cells far away from an equipped laboratory and in nonsterile conditions, the freeze-drying method has been developed and tested in practice. This method was previously described for *E. coli* cells by the ATCC organization (the American Type Culture Collection, www.atcc.org (accessed on 20 August 2022)) and in some publications as well [36]. In the current study, we examined the freeze-drying procedure on the *E. coli* Δ*tolC* strain transformed with the pDualrep2 plasmid. As a result, we observed that freeze-dried cells retain reporter activity for at least eight and a half months (Figure 3). In addition, they can be stored both at room temperature and at 4 °C, which makes the stocks of freeze-dried reporter cells most suitable for application in the course of the Citizen Science project.

### 2.3. The Pipeline of the Citizen Science Project

The Citizen Science project inspired by the Ministry of Science and Higher Education of the Russian Federation (075-15-2021-1085) provides schoolchildren, students, teachers and the interested public throughout Russian regions a unique opportunity to work with scientists as part of the global initiative to discover new antibiotics from soil actinobacteria.

The first and most essential stage of the project is to choose the environmental locus from which sampling will be carried out. It can be a poorly studied natural niche, farmland, urban landscape or hard-to-reach area. Volunteers collect soil or sediment samples in sterile containers (Figure 4A) and bring them to a school laboratory, educational center or home. All information about the selected location, description of the territory, relief, vegetation, etc., is recorded on a special website.

Using an isolation kit containing sterile plates and reagents, the volunteers prepare serial 10-fold dilutions of collected samples and spread 50 μL of each suspension on isolation agar plates supplemented with antibiotics that limit the growth of Gram-negative bacteria and fungi. The inoculated plates should be wrapped with Parafilm^®^ M and incubated either in a thermostat at 28 °C or at room temperature for 2–3 weeks until noticeable colonies of actinobacteria appear, which can be recognized by a fluffy, velvety or leathery surface (Figure 4B).

Individual colonies are picked according to their cultural properties and restreaked by sterile toothpicks onto the surface of two ISP3 agar plates (Appendix A) [37]: one for further experiments and the other one for transfer to the Research Center. After 10–14 days of cultivation, isolates from one ISP3 plate are restreaked on different solid growth media known to promote the synthesis of natural products by actinomycete strains (Figure 4C, Appendix A). No pharmaceutical antibiotics are added to cultivation media at this stage. Acting carefully near an alcohol lamp or gas burner, the citizen scientist can streak 4–5 isolates on one plate, arranging them by sectors. The second ISP3 plate with actinobacteria lawns is transferred to the Research Center.

After another 10–14 days of cultivation, agar blocks with mycelium are cut out of actinobacteria lawns and placed on the surface of a solid LB medium coated with a reporter strain. One or two days later, the citizen scientist can observe not only zones of inhibition where reporter bacteria have not grown enough to be visible but also a characteristic response (reporter induction) that provides information about the mode of action of secondary metabolites with antibiotic properties (Figure 4D). At all stages, the volunteers record the results: they mark and describe isolates, photograph plates with colonies, streaks and reporter tests, and enter the information on the project website. Sending out Petri dishes with active antagonists to the Research Center completes the block of citizen science.

In the Research Center laboratory, the isolates are restreaked on fresh solid growth media, their purity is confirmed and activity is verified on the reporter strains. Thereafter, the isolated actinobacteria are cultivated in different liquid media under various conditions with the aim of identifying the optimal protocol with the highest yield of bioactive natural products (Figure 4E). Fermentation broths retaining activity in scaled-up cultivation are then subjected to solid-phase extraction on LPS-500-H sorbent. After initial sorption on the column, it is eluted stepwise with an ascending concentration of acetonitrile in water (Figure 4F). This approach makes it possible to combine the extraction and rough fractionation steps, providing active concentrates suitable for further HPLC analysis. In order to assess the stability of active components in acidic conditions, samples are treated with 0.1% trifluoroacetic acid (TFA) and tested for retention of antibiotic activity. The active fractions are then subjected to semi-preparative HPLC procedure, with acetonitrile and water being used as eluents under neutral or acidic conditions (Figure 4G). Newly collected fractions are concentrated in vacuo and tested on the reporter strains. For each analyte, HPLC conditions are repeatedly refined to identify the exact localization of the active substance. Then, the active pure peak is collected and analyzed with high-resolution mass spectrometry (HRMS) in positive and negative ionization modes with fragmentation of the three most abundant ions (Figure 4H). Mass spectra are processed manually with deconvolution and adduct analysis.

The resulting dataset on the active compound includes: (a) UV-Vis spectrum; (b) HPLC retention time; (c) putative mechanism of action, inferred from experiments with the reporter strains; (d) the exact mass of the substance, supported by the analysis of the adduct pattern in the mass spectra; (e) fragmentation spectra of the most intense ions in both positive and negative modes. The resulting dataset is used to identify the compound, using various databases of natural products [38]. If the purified active substance has not been previously described in the literature data, then it is further subjected to NMR spectroscopy to reveal its molecular structure.

Optionally, the pipeline can be supplemented with an additional analysis of fractions on a panel of bacterial strains specifically resistant to known antibiotics. Such an option may sometimes help to avoid wasting time on “rediscovering” already known compounds.

### 2.4. Phenotypic, Phylogenetic and Physiological Characteristics of Two New Producing Actinobacteria Strains

In the course of the Citizen Science project, two strains, KB-1 and BV113, were isolated from the urban soil of Moscow and moss from Sochi, respectively. Both strains demonstrated prominent antibiotic activity in tests on the reporter cells (Appendix A). KB-1 exhibited strong pTrpL2A-Katushka2S reporter induction, indicating that the active compound produced by the isolate functions as an inhibitor of protein biosynthesis. Whereas BV113 exhibited the induction of pSulA-RFP, which points to its ability to produce a substance that elicits the SOS response in bacteria nearby. Using phenotypic features, the KB-1 and BV113 strains were identified as mycelial actinobacteria (Figure 5, Appendix A).

Comparative analysis of 16S rRNA sequences of KB-1 and BV113 strains with representatives of the family Streptomycetaceae confirmed that the isolates are closely related to species of the genus *Streptomyces*. The Maximum Likelihood tree (Figure 6) based on 16S rRNA gene sequences indicated that KB-1 forms a tight cluster with strains *S. zaomyceticus* NRRL B-2038^T^, *S. exfoliatus* NBRC 13191^T^, *S. venezuelae* ATCC 10712^T^ and *S. viridobrunneus* NBRC 15902^T^ (100% sequence similarity). The ability to synthesize pikromycin was noted earlier in *S. zaomyceticus* [39] and *S. venezuelae* ATCC 15439 [40]. High levels of 16S rRNA gene sequence similarity were also found between the strain BV113 and a group of *S. osmaniensis* OU-63^T^, *S. longwoodensis* NBRC 14251^T^ and *S. galbus* JCM 4570^T^ (99.7%), which constitute a well-supported cluster on the phylogenetic tree, with a 91% bootstrap value (Figure 6). The isolate BV113 was also found to share relatively high 16S rRNA gene similarity with *S. chartreusis* JCM 4570^T^ (99.2%), known as a producer of chartreusin [41].

The position of strains KB-1 and BV113 in the phylogenetic tree was unaffected by the choice of tree-making algorithm or outgroup strains used.

The physiological and biochemical characteristics of KB-1 and BV113 strains compared with closely related strains are given in Appendix A.

### 2.5. Identification of Active Compounds Produced by Streptomyces sp. KB-1 and BV113 Strains

The first object of the study was *Streptomyces* sp. KB-1, which demonstrated inhibition of protein biosynthesis on the reporter strains (Appendix A). Its culture liquid was subjected to solid-phase extraction, with an active fraction being eluted from the LPS-500-H sorbent with a 20% aqueous acetonitrile (MeCN). Activity-guided HPLC analysis revealed a specific active metabolite with a maximum UV absorption at 274 nm (Figure 7). The mass spectrum of this compound contained the main adduct [M+H]^+^ with *m*/*z* value 526.3370, exhibiting a sole fragment ion observed at *m*/*z* 158.1184 in the MS/MS spectrum (Appendix A). The exact mass of the detected metabolite corresponds to the composition C_28_H_47_NO_8_ (the calculated *m*/*z* value for [M+H]^+^, 526.3374).

The search for candidates was carried out based on the accumulated data using the NPAtlas, Dictionary of Natural Products and PubChem databases. The characteristic fragmentation allowed us to conclude that the active compound is a known inhibitor of protein biosynthesis, pikromycin (Figure 7) [42,43].

The second object of the study was *Streptomyces* sp. BV113, which demonstrated the induction of SOS response in tests on the reporter strains (Appendix A). Solid-phase extraction of the *Streptomyces* sp. BV113 strain fermentation broth using the LPS-500-H sorbent yielded an active fraction eluted with 30% acetonitrile in water. HPLC analysis of the fraction (Figure 8) showed that the activity was associated with a hydrophobic compound with characteristic long wavelength maxima in the UV-Vis spectrum (400, 422 nm). Mass spectrometric analysis of this compound revealed a prominent [M+NH_4_]^+^ ion peak at *m*/*z* 658.213 in positive ion mode, corresponding to the molecular formula C_32_H_32_O_14_ (the calculated *m*/*z* value for [M+NH_4_]^+^, 658.2130).

Glycosylated benzochromenone chartreusin (Figure 8) was selected as the most appropriate candidate. The structural hypothesis was confirmed by comparing the fragmentation of this compound with the literature data. Fragmentation of the [M+Na]^+^ ion with *m*/*z* value 663.1677 contains key fragment ions with *m*/*z* values 329.1207 and 503.0943 (Appendix A), which correspond to the loss of glycosidic fragments of the molecule [44]. Consistent with the results of the reporter strains test, chartreusin is known to bind to GC-rich tracts in DNA and cause single-strand breaks [45].

## 3. Materials and Methods

### 3.1. Reporter Strains and Medium

*E. coli* JW5503 strain with deletion of the Δ*tolC* gene (referred to here as *E. coli* Δ*tolC*) was kindly provided by Hironori Niki, National Institute of Genetics, Japan [46]. *E. coli* BW25113 strain with partial deletion of the *lptD* gene, codons 330 to 352, (referred to here as *E. coli*
*lptD*) was kindly provided by Alexander S. Mankin, University of Illinois, Chicago, IL, USA [35]. Both strains were transformed with either new reporter plasmids or a double reporter system pDualrep2 [24].

*E. coli* strains were grown at 37 °C in LB medium supplied with 100 μg/mL ampicillin, if required.

### 3.2. Plasmids and Cloning

To create the construct pDualrep3, the vector backbone was amplified by high-fidelity PCR from the pDualrep2 plasmid [24] using primers 5′-CCAGCACAGTGGTCGAAG-3′ and 5′-CATATGTTGTGTTTGCATTGTTATTCTC-3′. The *lacZ* gene was amplified by PCR from the pJC27 plasmid [47] with 5′-CAATGCAAACACAACATATGACCATGATTACGCCAAGC-3′ forward and 5′-TTCTTCGACCACTGTGCTGGAATACGGGCAGACATGGC-3′ reverse primers. The joining of two DNA fragments was performed with the NEBuilder^®^ HiFi DNA Assembly technique (NEB).

The plasmid pTrpL2A-Katushka2S was obtained by PCR amplification with primers 5′-GGGCCCGCGACTCTAGATCATAATCA-3′ and 5′-GGTTCAGTAGAAAAGATCAAAGGATC-3′ using pDualrep2 as a template followed by blunt-end DNA ligation.

The plasmid pTrpL2A-RFP was obtained from pTrpL2A-Katushka2S by replacing the *katushka2S* gene with *turbo-rfp*. The vector was amplified by PCR with primers 5′-CCAGCACAGTGGTCGAAG-3′ and 5′-TTCTCCTTGATCAGCTCGCCCATATGTTGTGTTTGCATTGTTATTCTC-3′. The *turbo-rfp* gene was amplified from the pDualrep2 plasmid with primers 5′-GGCGAGCTGATCAAGGAG-3′ and 5′-TTCTTCGACCACTGTGCTGGAAGCTTGTCGACCTGCAG-3′. The joining of two DNA fragments was performed using the NEBuilder^®^ HiFi DNA Assembly technique (NEB).

The reporter construct pSulA-RFP was obtained from the pDualrep2 plasmid by PCR amplification with primers 5′-CCAGCACAGTGGTCGAAG-3′ and 5′-TTCTTCGACCACTGTGCTGGAAGCTTGTCGACCTGCAG-3′ and subsequent NEBuilder^®^ HiFi DNA Assembly technique (NEB).

The *E. coli* JM109 strain was used for DNA cloning. Sequences of intermediate products and final constructs were confirmed by sequencing with appropriate primers. Plasmid maps were visualized using the program SnapGene^®^ Viewer (version 5.2.4).

### 3.3. Reporter Cells Freeze-Drying and Storage

For freeze-drying, an overnight culture (OD_600_ 0.9–1.0) of *E. coli* Δ*tolC* transformed with the pDualrep2 plasmid was used. In total, 500 µL of the overnight culture was transferred into sterile 2.0 mL centrifuge tubes, cells were harvested by centrifugation at 7000 rpm for 2 min (~8400× *g*, Centrifuge 5418 R, Eppendorf, Hamburg, Germany), washed and resuspended in 500 µL of the lyophilization medium: 1% gelatin (*w*/*v*), 1% monosodium glutamate (*w*/*v*), 10% sucrose (*w*/*v*) and distilled H_2_O. Prepared lyophilization medium was sterilized by filtration through a 0.45 µm filter prior to use. 

Opened centrifuge tubes with the suspension of reporter cells were covered with Parafilm^®^ M. Thereafter, tubes were subjected to gradual freezing: 20 min at 4 °C, then freezing in liquid nitrogen. Drying of samples was carried out at a pressure below 0.370 mBar for 24 h (FreeZone Plus 2.5 Liter Cascade Benchtop Freeze Dry System, Labconco, Kansas City, MO, USA). The parafilm was removed from dried samples; half of them were stored at room temperature and the other half at 4 °C. On the day of the experiment, freeze-dried stocks were restored by adding 2 mL of fresh sterile LB medium.

### 3.4. Reporter Assays on Agar Plates

The overnight cultures of reporter cells diluted 5–10 times with fresh sterile LB medium or the restored freeze-dried stocks were plated on LB agar medium supplied with 100 μg/mL ampicillin. The test samples were placed on the surface of dried agar plates covered with a reporter strain. For reporter activity validation tests, 1.5 µL of the following antibiotics were used: chloramphenicol (1 mg/mL), puromycin (2 mg/mL), erythromycin (5 mg/mL), doxorubicin (2 mg/mL), spectinomycin (5 mg/mL), streptomycin (5 mg/mL), fusidic acid (5 mg/mL), clindamycin (10 mM), lincomycin (10 mM), levofloxacin (30 µg/mL), kanamycin (5 mg/mL) and ampicillin (100 mg/mL); as well, disks soaked in antibiotic solution were used for tetracycline (30 µg) and rifampicin (5 µg).

In order to perform reporter activity tests with actinobacteria isolates grown on different nutrient media, agar blocks with mycelium were cut out of the grown lawns using the wide end of sterile 1000 µL pipette tips and were placed on the surface of dried plates coated with a reporter strain.

To test the reporter and antibacterial activity of liquid samples, such as culture liquids or fractions obtained after solid-phase extraction or HPLC analysis, dried plates covered with a reporter strain were subjected to cutting wells out of the agar medium using the wide end of sterile 1000 µL pipette tips. The free volume of the resulting wells was 100 µL. For culture liquids and HPLC fractions, 100 µL of solution per well was used; for solid-phase extraction eluates, 10 µL of solution mixed with 90 µL of distilled H_2_O per well was used in the agar diffusion assay.

Following overnight incubation at 37 °C or at room temperature (RT) for 1–2 days, the Petri dishes were photographed or, if possible, scanned by a ChemiDoc™ Imaging System (Bio-Rad, Hercules, CA, USA) with two channels: Cy3 (emission filter 605 ± 50 nm, green pseudocolor) for TurboRFP fluorescence and Cy5 (emission filter 695 ± 50 nm, red pseudocolor) for Katushka2S fluorescence.

### 3.5. Sampling and Isolation of Actinobacteria

Soils, sea sediments and plant samples were collected from different regions of Russia during the summer of 2021. The samples were placed into sterile containers to prevent contamination, delivered to an educational laboratory and stored at 4 °C until investigation. Sample solutions were prepared by dissolving 1 g of materials in 99 mL distilled water. To facilitate dissolution, sample flasks were shaken at 200 rpm for 10 min (Innova^®^ 44 Shaker, New Brunswick Scientific, Edison, NJ, USA). Aliquots of serial 10-fold dilutions were spread on isolation media: mineral agar gauze 1 [48], organic medium 79 [49], M490 [50] and HV agar [51] (Appendix A) supplemented with nystatin (250 μg/mL) and nalidixic acid (10 μg/mL) to prevent the growth of fungi and Gram-negative bacteria, respectively. After 14–21 days of incubation at 28 °C or at room temperature (RT), the powdery-surfaced and leathery colonies were recognized as actinobacteria strains, picked, restreaked and grown on two fresh ISP3 agar plates (Appendix A) [37]: one for further experiments and the other one for transfer to the Research Center.

Thereafter, individual actinobacteria isolates were restreaked and grown on a set of nutrient media: mineral agar gauze 1 (G1), organic medium 79 (Org79), glucose-asparagine agar (GA), soy flour mannitol agar (SFM) and oatmeal agar (ISP3) for 10 days at 28 °C or at RT (Appendix A). Lawns of isolated strains were subsequently subjected to the agar block diffusion screening for antibiotic activity against reporter strains.

In the Research Center, the isolated actinobacteria strains were kept on oatmeal agar (ISP3) slants and stored as suspensions of spores in 20% glycerol (*v*/*v*) at −20 °C.

### 3.6. Phenotypic, Morphological and Physiological Characterization of New Producing Actinobacteria Strains

Aerial spore-mass color, substrate mycelial pigmentation, the production of diffusible pigments and melanin were recorded after incubation at 28 °C for 14 days on media recommended by the International Streptomyces Project (ISP) [37]. Morphological characteristics of aerial hyphae of KB-1 and BV113 strains were analyzed after 2 weeks of incubation at 28 °C on ISP4 with CamScan S2 scanning electron microscope (Cambridge Instruments, Cambridge, UK) and on ISP3 medium with JSM-6380LA scanning electron microscope (JEOL Ltd., Akishima, Tokyo, Japan), respectively.

Carbon source utilization was assessed after incubation at 28 °C for 14 days on basal agar medium (ISP9) (Appendix A) [37] supplemented with bromocresol purple indicator solution, 0.04% (*w*/*v*). Cellulose decomposition, starch hydrolysis, nitrate reduction, milk peptonization, gelatin liquefaction and H_2_S production were examined as described previously [52].

### 3.7. Phylogenetic Analysis of New Producing Actinobacteria Strains

Genomic DNA extraction from actinobacteria isolates was carried out according to the procedures described previously [53]. Amplification of the 16S rRNA gene was performed by high-fidelity PCR using primers 5′-GGATGAGCCCGCGGCCTA-3′ (243F) and 5′-CCAGCCCCACCTTCGAC-3′ (A3R) [28]. The sequencing was conducted with ABI Prism^®^ BigDye™ Terminator v3.1 Cycle Sequencing Kit (Applied Biosystems, Waltham, MA, USA) and detected with Applied Biosystems^®^ 3730 DNA Analyzer (Life Technologies, Carlsbad, CA, USA) in the Center for Collective Use “Genome” (Moscow, Russia). The contigs were processed and assembled with the GeneStudio™ Pro software (Version 2.2.0.0). GenBank accession numbers for the 16S rRNA gene sequences of *Streptomyces* sp. KB-1 and BV113 strains are OM780307.1 and OM801975.1, respectively.

Taxonomic affiliation of the isolates was assessed using the 16S rRNA gene sequences as a query in the BLAST (https://blast.ncbi.nlm.nih.gov/Blast.cgi (accessed on 6 March 2022)) and EzTaxon (www.ezbiocloud.net (accessed on 6 March 2022)) web services. The 16S rRNA gene sequences of the most closely related species of the genus *Streptomyces* (more than 99% of identity) were used to build the alignment. In total, it comprised 25 nucleotide sequences, with *Amycolatopsis rifamycinica* DSM 46095^T^ as an outgroup. The alignment was manually trimmed to 1147 bp and the phylogenetic tree was reconstructed by the Maximum Likelihood (ML) method based on the Tamura–Nei model [54], as well as by the Neighbor-Joining (NJ) method [55] based on the Kimura two-parameter model [56]. The phylogenetic tree was visualized using MEGA 7.0 software [57].

### 3.8. Cultivation and Extraction of SECONDARY metabolites

Actinobacteria isolates were first inoculated in 20 mL of liquid organic medium 79 (Org79, no agar) and cultivated at 28 °C with constant shaking (180 rpm, Innova^®^ 44 Shaker, New Brunswick Scientific, Edison, NJ, USA) for 3 days. The resulting starters were used to inoculate 200 mL of different liquid media (Appendix A) and then suspensions were incubated at 28 °C with constant shaking (220 rpm, Innova^®^ 44 Shaker, New Brunswick Scientific, Edison, NJ, USA) in 750 mL Erlenmeyer flasks to determine the optimal cultivation conditions.

For *Streptomyces* sp. KB-1, the greatest production of an antibacterial metabolite was observed upon cultivation in a modified ISP3 liquid medium for 6 days. Composition of the modified ISP3 medium (g/L) was as follows: oatmeal—20, powder chalk—5, FeSO_4_·7H2O—0.001, MnCl_2_·4H_2_O—0.001, ZnSO_4_·7H_2_O—0.001, pH 7.2.

For *Streptomyces* sp. BV113, the greatest production of an antibacterial metabolite was observed upon cultivation in a modified Org79 liquid medium for 4 days. Composition of the modified Org79 medium (g/L) was as follows: maltose—10, peptone—10, casein hydrolysate—2, yeast extract—2, NaCl—6, pH 7.2.

Culture liquids were separated from biomass by centrifugation at 20,000× *g* for 5 min (Centrifuge 5810 R, Rotor FA-45-6-30, Eppendorf, Hamburg, Germany). The supernatants were subjected to solid-phase extraction and primary fractionation on LPS-500-H sorbent (LLC “Technosorbent”, Moscow, Russia) using water-acetonitrile mixtures as eluents.

### 3.9. Antibiotic Identification

HPLC analysis and fractionation were performed with the Vanquish Flex UHPLC System using the Diode Array Detector (Thermo Fisher Scientific, Waltham, MA, USA), equipped with Luna^®^ 5 µm C18(2) 100 Å, 250 × 4.6 mm column (Phenomenex, Torrance, CA, USA).

Mass spectra were collected using maXis II 4G ETD mass spectrometer (Bruker Daltonics, Bremen, Germany) and UltiMate 3000 chromatograph (Thermo Fisher Scientific, Waltham, MA, USA), equipped with Acclaim RSLC 120 C18 2.2 µm 2.1 × 100 mm column (Thermo Fisher Scientific, Waltham, MA, USA). Spectrum registration mode: ESI ionization mode, full scan from 100–1500 *m*/*z*, MS/MS with selection of the three most intense ions, dissociation type: CID 10–40 eV, nitrogen collision gas. Mass spectra were processed using OpenChrom Lablicate Edition (1.4.0.202201211106), TOPPView v. 2.6.0 [58]. The chemical structures were identified using the GNPS [59], NPAtlas [60,61] and Dictionary of Natural Products 31.1 (https://dnp.chemnetbase.com (accessed on 10 March 2022)) [38] databases.

## 4. Conclusions

In order to speed up the screening for new antibiotic-producing strains, we created reporter plasmids that can be utilized in the course of the Citizen Science project. The ability to detect reporter induction with the naked eye makes it possible to crowdsource not only sampling but also some primary experiments such as isolation of single colonies and testing them against indicator reporter strains. Consistent with the idea of giving citizen scientists an opportunity to collect and analyze natural samples near their homes, we developed a special optimized pipeline that has been successfully applied in practice. Two examples of antibiotic-producing strains discovered de novo are presented in the article: *Streptomyces* sp. KB-1 (produces pikromycin) and BV113 (produces chartreusin). Through a proof of concept, we demonstrate that the novel workflow of the Citizen Science project is an effective means to improve the screening productivity.

## Figures and Tables

**Figure 1 antibiotics-11-01198-f001:**
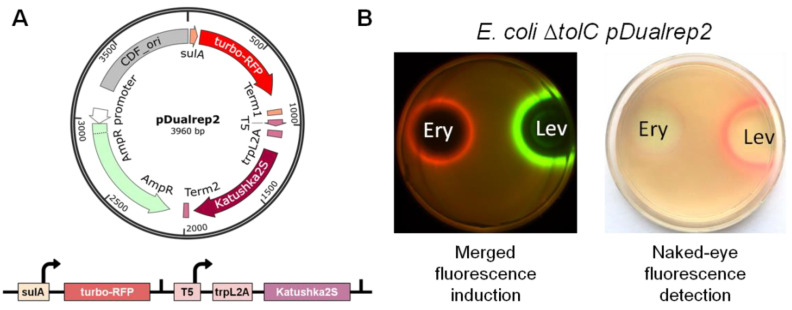
Double reporter system pDualrep2. (**A**) pDualrep2 plasmid map and the reporter scheme. CDF_ori, CloDF13-derived CDF replicon; sulA, promoter of the *sulA* gene; T5, bacteriophage T5 promoter; trpL2A, modified *trpL* leader open reading frame carrying W10A and W11A substitutions. Transcription start sites are shown by arrows. Transcription terminators are shown by vertical dashes. (**B**) Comparison of fluorescence signals using an imaging system (left) and with the naked eye (right). An agar plate was coated with the *E. coli* Δ*tolC* strain transformed with the pDualrep2 plasmid and spotted with erythromycin (Ery) and levofloxacin (Lev). The plate was scanned in Cy3 (for TurboRFP) and Cy5 (for Katushka2S) channels, shown as green and red pseudocolors, respectively.

**Figure 2 antibiotics-11-01198-f002:**
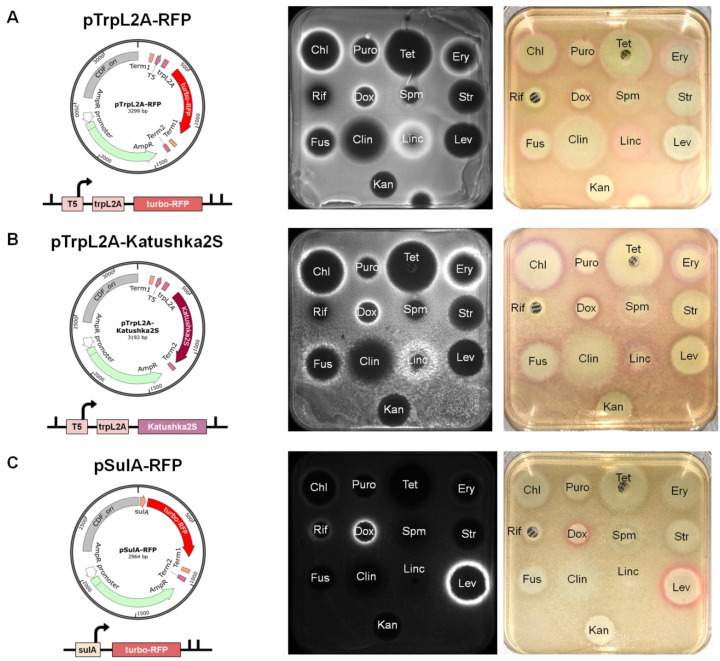
New reporter constructs pTrpL2A-RFP (**A**), pTrpL2A-Katushka2S (**B**) and pSulA-RFP (**C**) were created, transformed into the *E. coli* Δ*tolC* strain and validated with a panel of antibiotics. The plates (**A**,**C**) were scanned in the Cy3 (TurboRFP) channel; the plate (**B**) was scanned in the Cy5 (Katushka2S) channel (middle). TurboRFP and Katushka2S are visible to the naked eye in orange-red and lilac colors, respectively, in the zone of antibiotic sublethal concentrations (right). The panel of antibiotics is as follows: chloramphenicol (Chl), puromycin (Puro), tetracycline (Tet), erythromycin (Ery), rifampicin (Rif), doxorubicin (Dox), spectinomycin (Spm), streptomycin (Str), fusidic acid (Fus), clindamycin (Clin), lincomycin (Linc), levofloxacin (Lev) and kanamycin (Kan).

**Figure 3 antibiotics-11-01198-f003:**
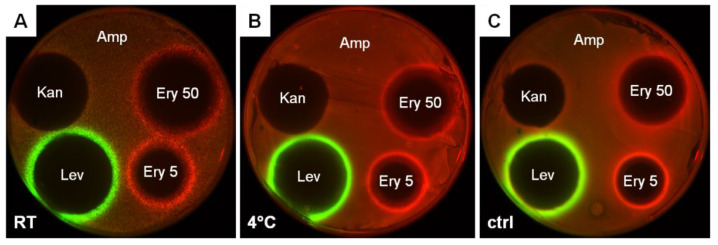
Comparison of fluorescence induction after the freeze-dried cells were kept for 8.5 months (257 days) at room temperature (**A**), at 4 °C (**B**) and overnight culture as a control (**C**). Agar plates were coated with the *E. coli* Δ*tolC* strain transformed with the pDualrep2 plasmid and spotted with antibiotics at the following concentrations: ampicillin (Amp, 100 mg/mL), erythromycin at two concentrations (Ery 50, 50 mg/mL, and Ery 5, 5 mg/mL), levofloxacin (Lev, 25 µg/mL) and kanamycin (Kan, 50 mg/mL). The plates were scanned in Cy3 (TurboRFP) and Cy5 (Katushka2S) channels, shown as green and red pseudocolors, respectively.

**Figure 4 antibiotics-11-01198-f004:**
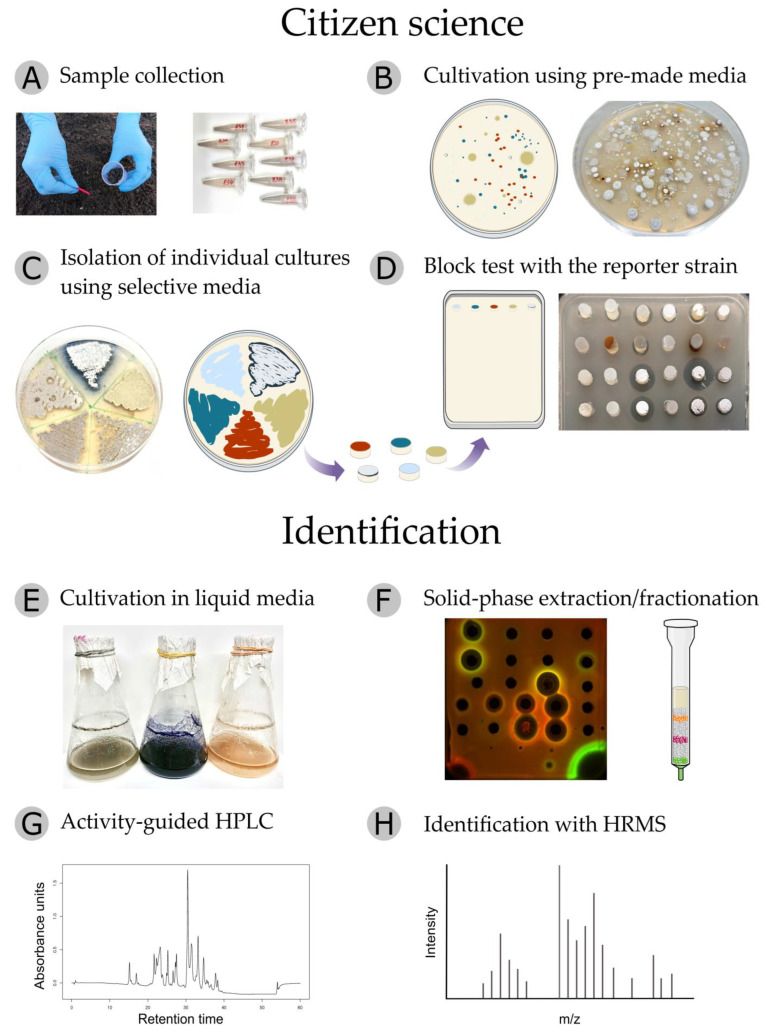
Workflow diagram of the Citizen Science project. (**A**) Soil sampling. (**B**) Isolation of individual actinomycete colonies through serial dilution. (**C**) Pure isolates are streaked onto different selective growth media. One extra plate is transferred to the Research Center. (**D**) Activity-guided selection of antimicrobial producers using agar block diffusion assay with the reporter strains. (**E**) Determination of optimal cultivation conditions. (**F**) Solid-phase extraction of culture liquid followed by fractions analysis using agar-well diffusion assay with the reporter strains. (**G**) Bioactive fractions are repeatedly subjected to reversed-phase HPLC separation. (**H**) Active pure HPLC peaks are analyzed with HRMS, revealing the exact mass of the antimicrobial compound.

**Figure 5 antibiotics-11-01198-f005:**
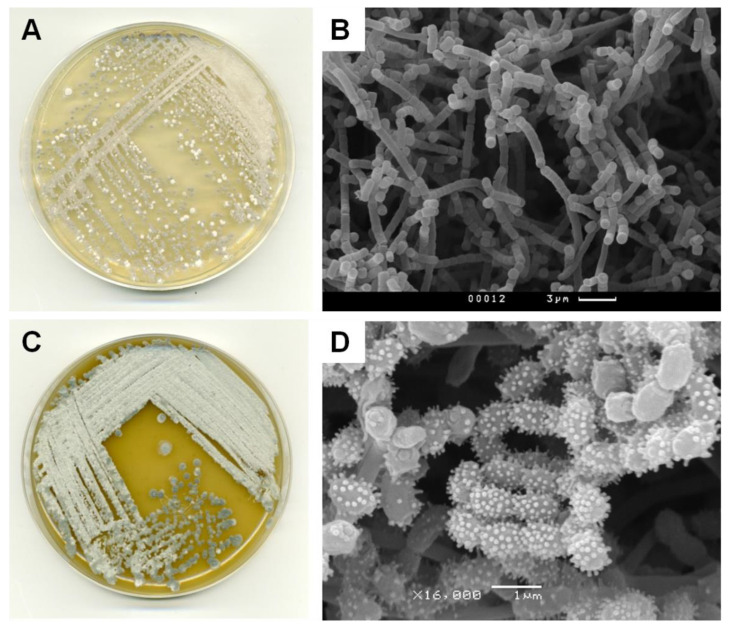
Morphological properties of *Streptomyces* sp. KB-1, grown on ISP4 medium at 28 °C for 14 days (**A**,**B**), and *Streptomyces* sp. BV113, grown on ISP3 medium at 28 °C for 14 days (**C**,**D**). (**A**,**C**) Photographs of plates with strain streaks. (**B**,**D**) Mycelium micrographs taken on a scanning electron microscope (SEM).

**Figure 6 antibiotics-11-01198-f006:**
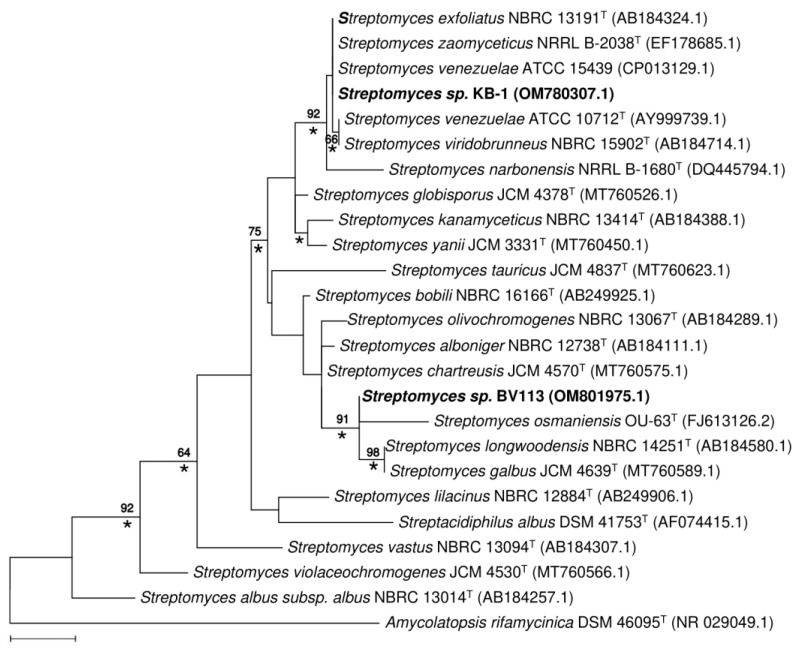
Maximum Likelihood phylogenetic tree based on the Tamura–Nei model, showing the relationship between isolated strains, KB-1 and BV113, and representative members of the genus *Streptomyces* (based on 1147 unambiguously aligned nucleotides of 16S rRNA gene sequence). Numbers at nodes are bootstrap support percentages based on 1000 sampled datasets; only values above 60% are shown. Asterisks (*) indicate the branches that were also found in the Neighbor-Joining tree. Bar, 0.01 substitutions per nucleotide position.

**Figure 7 antibiotics-11-01198-f007:**
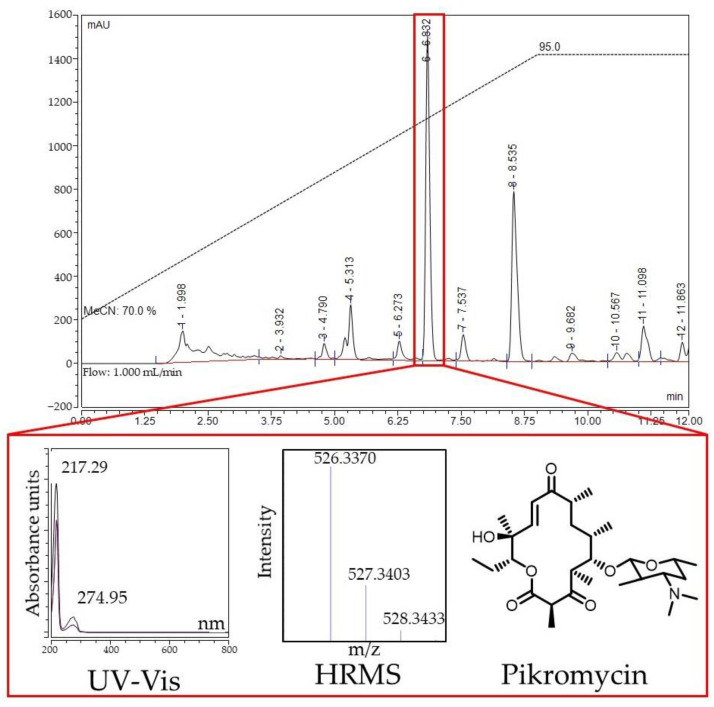
HPLC analysis of the active fraction of *Streptomyces* sp. KB-1 culture liquid. Elution with 70→95% MeCN in water for 9 min followed by 3 min of 95% MeCN. Red box indicates the active component on the HPLC profile.

**Figure 8 antibiotics-11-01198-f008:**
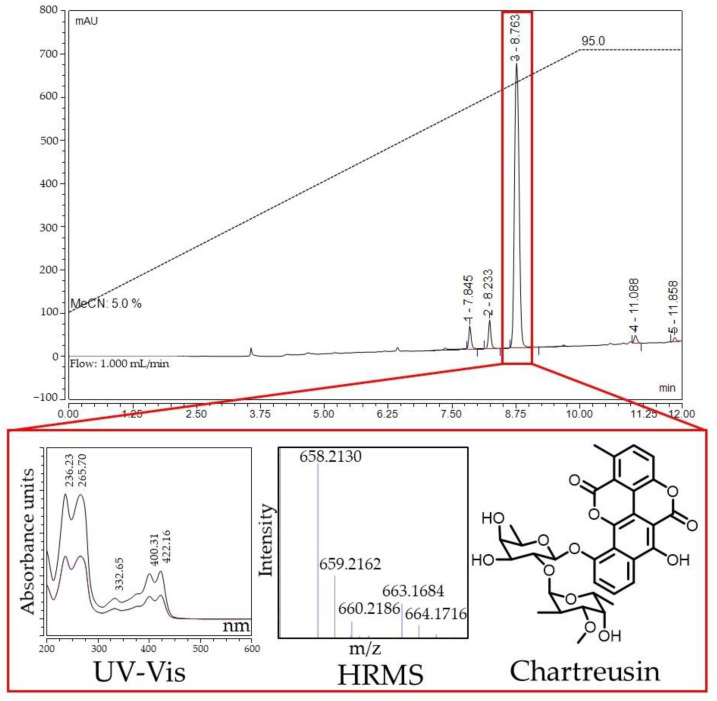
HPLC analysis of the active fraction of *Streptomyces* sp. BV113 culture liquid. Elution with 5→95% MeCN in water for 10 min followed by 2 min of 95% MeCN. Red box indicates the active component on the HPLC profile.

## Data Availability

Not applicable.

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
