# Peer review of "Mechanism-Based Approach to New Antibiotic Producers Screening among Actinomycetes in the Course of the Citizen Science Project"

_antibiotics, 2022, doi:10.3390/antibiotics11091198_

Round 1

Reviewer 1 Report

This study involves a survey of antibiotic production in soil microorganisms with a sort of crowdsourcing applied to the sample collection and initial culturing phases of the process. While it is a very resourceful and straightforward approach, the difficulties in the presentation with language and grammar hamper any real clarity of analysis by the reader. The  improved visual reporter assay for use by citizen scientists is excellent, the figures and some of the method descriptions are good, and the supplemental appears appropriate. Perhaps more emphasis on this being a proof-of-concept study would be helpful as well as discussion of its application to finding soil producers of novel antibiotic agents. I had attempted a rewording of the abstract as part of understanding the scope and focus of the study and I have attached it. 

Author Response

We are thankful to the reviewer for such a careful and critical reading of the manuscript and valuable comments.

Point 1: This study involves a survey of antibiotic production in soil microorganisms with a sort of crowdsourcing applied to the sample collection and initial culturing phases of the process. While it is a very resourceful and straightforward approach, the difficulties in the presentation with language and grammar hamper any real clarity of analysis by the reader.

Response 1: We would like to thank the reviewer for noting the problems with language and grammar. We have tried to improve the quality of the language and grammar throughout the manuscript. Most ambiguous sentences have been reworded. Hope, now the manuscript is more pleasant to read. If it is still not correct enough, please, let us know and we will submit the manuscript to English Editing procedure.

Point 2:  Perhaps more emphasis on this being a proof-of-concept study would be helpful as well as discussion of its application to finding soil producers of novel antibiotic agents.

Response 2: We have attempted to make an emphasis on a "proof-of-concept" approach, mainly in the introduction and conclusion sections. We have expanded the content of the introduction, added more information about other Citizen Science projects aimed at crowdsourcing the search for new antibiotics. Moreover, we added a block about other approaches used to increase the discovery rates of new antibiotics. Thus, we explained what is the Citizen Science project and how it should be applied in practice.

Point 3: I had attempted a rewording of the abstract as part of understanding the scope and focus of the study and I have attached it.

Response 3: Thank you for rewording the abstract. We have taken advantage of all your corrections. We hope that now the abstract, as well as the rest of the manuscript, has become more pleasant and understandable to read.

Reviewer 2 Report

In this study, Volynkina et al. demonstrated several reporters containing eye-visible fluorescent protein genes, which can be used to increase the efficiency of determining the mechanism of antibiotics at the very initial stage of screening. The combination of mechanism-based approaches and civil science were adopted to prove the effectiveness in practice, which reveals a significant increase in the screening rate. More valuably, two new strains Streptomyces sp. KB-1 and BV113 were found to produce antibiotics pikromycin and chartreusin, respectively, demonstrating the efficiency of the pipeline. Overall, this manuscript is well organized and the results look sound. I think it is worthy of publication after some detailed revisions to improve the quality of the work. I have few comments and questions for the content of the work:

Major points:

Introduction, the contents is too short. This section should discuss relationships of this study to previously published work, and state the significance, originality, or contribution to new knowledge concisely. Thus, this section should be supplemented considerably to attract the reader's interest and understanding. And more related literatures should be added.

Minor points:

1. Figure 4, the subheadings (1-8) in the Figure 4 are suggested to revise to (A-H).

2. Figure 4, for the sake of drawing, it is suggested to supplement the graph in subheading (8)

3. Page 5-7, the contents corresponding to Figure 3(1)(2)…(8) should the revised to Figure 4A, B…F.

4. Page 9, the sentence “… with a UV maximum at 274 nm” should be revised to “… with a maximum UV absorption at 274 nm”.

5. Page 9, “The mass spectrum of this compound … in the MS/MS spectrum”. Please supplement the corresponding MS/MS spectrum of [M+H]+ at m/z 526.3370 and the corresponding fragment ion at m/z 158.1184 in the Supplementary Materials.

6. Page 10, “Fragmentation of the [M+Na]+…glycosidic fragments of the molecule”. Please supplement the corresponding MS/MS spectrum of [M+Na]+ at m/z 663.1677 in the Supplementary Materials.

7. Page 9-10, the font of the word “pikromycin” and “chartreusin” should not be bold.

8. Figure 7 and 8, the graph resolution is relatively low, it is hard to observe the coordinate axis and annotation in the UV-Vis and HRMS graphs.

9. Reference, some scientific name should be italic, such as Mycobacterium tuberculosis (Ref. 8), Escherichia coli (Ref. 13), Streptomyces venezuelae (Ref. 16), Streptomyces chartreusis (Ref. 17), Escherichia coli (Ref. 22).

Author Response

We are thankful to the reviewer for such a careful and critical reading of the manuscript and valuable comments. 

Point 1: Introduction, the contents is too short. This section should discuss relationships of this study to previously published work, and state the significance, originality, or contribution to new knowledge concisely. Thus, this section should be supplemented considerably to attract the reader's interest and understanding. And more related literatures should be added.

Response 1: We have expanded the content of the introduction, added more information about other Citizen Science projects aimed at crowdsourcing the search for new antibiotics. Moreover, we have added a block about other approaches used to increase the discovery rates of new antibiotics. Now the introduction looks more attractive.

Point 2: Figure 4, the subheadings (1-8) in the Figure 4 are suggested to revise to (A-H). Figure 4, for the sake of drawing, it is suggested to supplement the graph in subheading (8)

Response 2: Thank you for the comment. We have corrected the Figure 4, and replaced the digits (1-8) with letters (A-H). In addition, we have added a graph in subheading (8) (now it is subheading H).

Point 3: Page 5-7, the contents corresponding to Figure 3(1)(2)…(8) should the revised to Figure 4A, B…F.

Response 3: Thank you for noticing the typo. We have corrected it.

Point 4: Page 9, the sentence “… with a UV maximum at 274 nm” should be revised to “… with a maximum UV absorption at 274 nm”.

Response 4: Thank you for correcting the grammar. In addition, we have tried to improve the quality of the language and grammar throughout the manuscript. Hope, now it is more pleasant to read.

Point 5: Page 9, “The mass spectrum of this compound … in the MS/MS spectrum”. Please supplement the corresponding MS/MS spectrum of [M+H]+ at m/z 526.3370 and the corresponding fragment ion at m/z 158.1184 in the Supplementary Materials.

Response 5: We have added the corresponding MS/MS spectrum in Supplementary Materials (Figure S5).

Point 6: Page 10, “Fragmentation of the [M+Na]+…glycosidic fragments of the molecule”. Please supplement the corresponding MS/MS spectrum of [M+Na]+ at m/z 663.1677 in the Supplementary Materials.

Response 6: We have added the corresponding MS/MS spectrum in Supplementary Materials (Figure S6).

Point 7: Page 9-10, the font of the word “pikromycin” and “chartreusin” should not be bold.

Response 7: We have corrected it.

Point 8: Figure 7 and 8, the graph resolution is relatively low, it is hard to observe the coordinate axis and annotation in the UV-Vis and HRMS graphs.

Response 8: We have replaced previous pictures with new ones. the resolution of the graphs in Figures 7 and 8 is much better.

Point 9: Reference, some scientific name should be italic, such as Mycobacterium tuberculosis (Ref. 8), Escherichia coli (Ref. 13), Streptomyces venezuelae (Ref. 16), Streptomyces chartreusis (Ref. 17), Escherichia coli (Ref. 22).

Response 9: We have corrected it. Now all strain names are written in italics in the list of references.

Round 2

Reviewer 2 Report

The authors have carefully addressed my concerns. Considering that the quality of this revised manuscript has been improved considerably, it can be accepted for publication as it is.